# Correlation between Language Development and Motor Skills, Physical Activity, and Leisure Time Behaviour in Preschool-Aged Children

**DOI:** 10.3390/children9030431

**Published:** 2022-03-18

**Authors:** Daniela Mulé, Ilka Jeger, Jörg Dötsch, Florian Breido, Nina Ferrari, Christine Joisten

**Affiliations:** 1Department for Physical Activity in Public Health, Institute of Movement and Neurosciences, German Sport University Cologne, 50933 Cologne, Germany; ilka.jeger@gmail.com (I.J.); f.breido@rfk.landeskrankenhaus.de (F.B.); 2Children’s Clinic and Polyclinic, University Hospital of Cologne, 50937 Cologne, Germany; joerg.doetsch@uk-koeln.de; 3Cologne Centre for Prevention in Childhood and Youth/Heart Centre Cologne, University Hospital of Cologne, 50937 Cologne, Germany; nina.ferrari@uk-koeln.de

**Keywords:** motor skills, inactivity, media consumption, overweight, SETK 3–5, KiMo-Test

## Abstract

Sedentary behaviour has a negative impact on children’s physical and mental health. However, limited data are available on language development. Therefore, this pilot study aimed to analyse the associations between language development and possible predictors such as motor skills and leisure time behaviour in preschool-aged children. Methods: In this cross-sectional analysis, motor skills and speech development status were assessed in 49 healthy preschool children. Physical activity and screen time were assessed via a parental questionnaire. Results: On average, physical activity was 8.2 ± 6.5 h/week; mean screen time was 154.2 ± 136.2 min/week. A positive relationship between the results in the item ‘One-leg stand’ and ‘Phonological working memory for nonwords’ (β-coefficient −0.513; *p* < 0.001) resp. ‘Formation of morphological rules’ (β-coefficient −0.626; *p* = 0.004) was shown within backward stepwise regression. ‘Lateral jumping’, resp. ‘Sit and Reach’ were positively associated with ‘Understanding sentences’ (β-coefficient 0.519; *p* = 0.001 resp. β-coefficient 0.735; *p* = 0.002). ‘Physical inactivity’ correlated negatively with all language development subtests (each *p* < 0.05). Media consumption had a negative predictive effect on the subdomain ‘Understanding Sentences’ (β-coefficient −0.530, *p* = 0.003). Conclusions: An inactive lifestyle correlated negatively with selected subtests of language development in early childhood. These results should be verified in larger groups and longitudinally but support the need for early health promotion.

## 1. Introduction

The modern lifestyle of children is characterised by increasing physical inactivity and media consumption [1,2,3,4,5,6,7]. Even at preschool age, most children fail to reach the nationally and internationally recommended physical activity levels. In Germany, for example, only 42.5% of preschool-aged girls and 48.9% of preschool-aged boys achieve the 180 min of physical activity per day recommended by the WHO (World Health Organisation) [2,8,9,10] and health experts in Germany [11]. Instead, 42% of the girls and 55% of the boys spend more than three hours a day during their early childhood years sitting still in front of screens [12]. This excessive amount of screen time, in turn, may negatively affect motor skills [13,14,15] and promote the development of body fat and visceral adipose tissue mass [16], leading to excess body weight and obesity [7].

A systematic review by Stiglic and Viner [17] showed that an increased screen time was associated with reduced cardiorespiratory fitness, increased obesity, less prosocial behaviour, lower self-esteem, depressive symptoms and lower quality of life [17]. Conversely, evidence indicates that physical activity in childhood improves fitness as well as motor skills [13,18,19,20,21], reduces the prevalence of obesity [22,23,24] and promotes physical and psychosocial development [25]. In addition, physical activity and fitness in childhood are associated with better cognitive abilities [26,27,28] and language development [29,30,31].

However, research has yet to determine whether motor skills in childhood, as a surrogate parameter for increased physical activity, are also associated with language development. Therefore, this pilot study analysed the language development status in preschool children and its association with motor skills and leisure time behaviour (physical activity, media use time), while taking socio-demographic factors and excess body weight into account.

## 2. Materials and Methods

### 2.1. Study Design

This cross-sectional study was approved by the Ethics Committee of the German Sport University Cologne (No. 018/21). Written informed consent was obtained from all parents before testing. In addition to the anthropometric measurements, the ‘kindergarten mobile test’ (author’s translation; ‘Kindergarten Mobil Test’ (KiMo) [32]) and the ‘speech development test for three- to five-year-old children’ (author’s translation; ‘Sprachentwicklungstest für drei-bis fünfjährige Kinder’ (SETK 3–5) [33]) were administered. A modified version of the standardised CHILT-III (‘Children’s Health Interventional Trial’) [34] questionnaire was employed to interview parents to get further data regarding their sociodemographic status, education and lifestyle as well as their children’s leisure time behaviour and potential predictive risk factors for children’s development (see below).

### 2.2. Sample

The sample was recruited of a randomly selected kindergarten and a ball sports group in North Rhine Westphalia. Healthy children aged 3 to 5 years were included. Exclusion criteria were severe physical or mental impairments (e.g., mutism), diagnosed speech and hearing disorders (e.g., cheiloschisis and palatoschisis, diagnosed developmental delays and hearing disorders), as well as acute illnesses or injuries. Of the 64 children examined, 49 healthy, typically developed children aged 3 to 5 years were included in the study (41% girls, 59% boys; see Figure 1).

### 2.3. Testing Procedure

#### 2.3.1. Parent’s Questionnaire

A modified version of the CHILT-III-questionnaire [34] was employed to interview parents regarding their own sociodemographic status, education and lifestyle as well as their children’s leisure time behaviour (e.g., daily and weekly physical activity and media use time). An inactive lifestyle was defined according to the National Association for Sport and Physical Education (NASPE) [35] (<60 min of structured/unstructured activity per day). Each child’s daily activity level, which the questionnaire recorded in minutes per day, was subdivided into general sporting activity (e.g., riding a pedal bike, unspecific playing with a ball or other sporting activities outside of club sports) and specific activity in club sports or kindergarten sports. For the final analysis, the corresponding activity times in the individual categories were summarised per week. For a better overview, the weekly physical activity time outside the nursery school and sports clubs has been summarised in hours. The age at which each child spoke his or her first word, as well as any medical abnormalities and possible risk factors for the child’s development, were recorded. Based on their years of schooling, parents were divided into a higher education level (>10 years of schooling) and a lower education level (<10 years of schooling) for the final analysis.

#### 2.3.2. Assessment of Anthropometric Data

For each child, current anthropometric data were recorded in a measurement protocol. Each child’s current age was determined from the date of birth and the date of the survey. The children’s weight and height were measured using Seca 761 scales and a Seca 225 calibrated rule. The children were measured and weighed in light clothing without shoes. BMI was calculated as body weight in kilograms divided by size in metres squared. The children were considered obese if their BMI exceeded the 97th percentile, being overweight if their BMI exceeded the 90th percentile but fell below the 97th percentile, normal weight if their BMI was greater than or equal to the 10th percentile but less than the 90th percentile and underweight if their BMI fell below the 10th percentile. The collected values were categorised in accordance with national reference data [36].

#### 2.3.3. Motor Skills Tests

This survey utilised the ‘kindergarten mobile test’ (KiMo) (author’s translation) [32]. All children completed the KiMo test in the morning in their kindergarten’s movement room under the supervision of an assigned test leader. The children wore sports shoes during all tests except the sit and reach tests, which they performed while wearing socks. The test leader explained and demonstrated all five test stations before asking the children to complete the corresponding activities.

(1) ‘Shuttle Run’ (SR) (Figure 2): This test determined the children’s speed and agility. Two squares (30 cm × 30 cm) were marked on the floor 4 m apart. Two wooden blocks (3.8 cm × 3.8 cm × 3.8 cm) were placed in one square. The children were asked to start in the square without wooden blocks and then carry the wooden blocks—one at a time—into the empty square, thus covering the distance between the squares four times. The test leader used a stopwatch to record the time (in seconds) each child required to complete this task.

(2) ‘Standing long jump’ (SLJ) (Figure 3): This test examined the explosive power in the children’s lower extremities. The children were asked to jump with both legs as far as possible from a starting line before landing on both legs without falling over. The distance was measured in centimetres, the better of two attempts was recorded for the study.

(3) ‘One-leg stand’ (OLS) (Figure 4): The children’s static balance and coordination were assessed on a 4.5 cm-wide wooden bar. The children were asked to stand as still as possible on one leg for 1 min. The number of times the free leg touched the ground was counted. The test was discontinued when the participants exceeded 30 floor contacts.

(4) ‘Sit-and-reach’ (SAR) (Figure 5): This test examined the flexibility of the hip joint and lumbar spine area as well as the capacity to stretch the ischiocrural muscles. The children were asked to sit with stretched knees in front of a wooden box (31 cm × 31 cm × 32 cm). The feet were placed flat against the box at a 90° angle. A tape measure was placed on the box. It was measured in centimetres how far the children could reach forward along the tape measure.

(5) ‘Lateral jumping’ (LJ) (Figure 6): This test determined the children’s coordination under time pressure and muscular endurance. The children were asked to stand on a wooden board (60 cm × 100 cm × 2 cm) and jump with both legs from side to side across a central bar (58 cm × 4 cm × 2 cm) as quickly as possible. Jumps that landed on the central line were not counted. The number of all jumps correctly performed in two rounds of 15 s each were counted.

#### 2.3.4. Speech Development Test for Children

The SETK 3–5 [33] is a standardised and norm-referenced instrument examining the language development of preschool children in Germany. Standardised on a group of 495 German-speaking children between 3.0 and 5.11 years of age, this test battery has exhibited high validity and reliability (with Cronbach’s alpha between 0.62 and 0.89). In particular, the SETK 3–5 assesses the domains of linguistic understanding, production and memory. The SETK 3–5 contains six tasks to measure the subsections of linguistic comprehension (‘Understanding sentences’ (US)), linguistic production (‘Formation of morphological rules’ (FMR), ‘Encoding semantic relations’ (ESR)) and linguistic memory (‘Phonological working memory for non-words’ (PMN), ‘Sentence memory’ (SM), ‘Memory span for word sequences’ (MSWS)). It includes two versions depending on the age of the children (one version for three-year-olds and another version for four- to five-year-olds). The following three subtests were included as they were completed by all participants:

The ‘understanding of sentences’ (US) subtest measures a child’s ability to comprehend sentences of varying complexity. In this subtest, the test leader read the three-year-olds a complete sentence and showed them a card with four different pictures. The children were then asked to select the picture that matched the sentence they had heard. In another subtest, the three- to five-year-old children sat in front of a series of objects and were given tasks related to these items, such as ‘Take the long pencil for yourself, and give me the short pencil.’ The total number of correctly performed tasks was recorded as the test score. A maximum score of 15 raw points was possible.

The ‘formation of morphological rules’ (FMR) subtest measures a child’s ability to build the plural forms of words. After hearing the singular form of a noun and seeing a corresponding picture card, the children were asked to form the respective plural. When administered to four- and five-year-old children, this test also included nonsense words. In addition, children could earn sub-points based on a list of deviating plurals defined in the test manual. Thus, a maximum total score of 36 raw points was possible.

The ‘phonological working memory for non-words’ (PMN) subtest measures a child’s ability to pronounce non-words. The children listened to the test leader saying nonsense names before being asked to repeat those names as accurately as possible. To increase the three-year-old children’s motivation, the test leader showed them colourful figures, which corresponded to the fantasy names. The test leader counted the number of nonsense names each child repeated correctly. A maximum raw score of 18 points was possible.

### 2.4. Data Analyses

The data analyses were performed with SPSS version 27.0. Descriptive data analyses included the calculation of arithmetic means, standard deviations (SD), minimum (Min.) and maximum (Max.) values. Comparisons of the independent variables were made using a *t*-test for independent samples. In addition, a multiple linear regression analysis was performed to assess the influence of socio-demographic factors and motor skills on the results of the language development subtests and the variables by stepwise backward elimination of non-significant variables. Therefore, the regression coefficient B, the β-coefficient and the coefficient of determination (corrected R-square) were determined. The initial model included the following variables: The raw scores of the KiMo test (SR (time in seconds), SLJ (distance in cm), OLS (number of ground contacts), SAR (distance in cm), LJ (number of correct jumps)), the risk factors (overweight, low maternal education level and physical inactivity), weekly activity time at the sports club (in minutes), physical activity time outside the sports club and kindergarten (in hours), weekly media consumption time, screen time and game use time on computers and game consoles (in minutes). Weekly physical activity time was divided into several categories to assess the effect of unstructured activity overall compared to structured activity in sports clubs and the effect of inactivity when children do not even reach the daily minimum recommended activity time. 

Significance was assumed at *p* < 0.05. The final linear regression analysis model was developed after stepwise removal of non-significant variables (inclusion at *p* < 0.1) by backward elimination.

## 3. Results

### 3.1. Anthropometric Data

The anthropometric data are presented in Table 1. Based on the BMI percentile standards of Kromeyer-Hauschild et al. [36], 8.2% of the children were overweight, 91.8% were of normal weight and no child was underweight.

### 3.2. Parent’s Questionnaire/Family Aspects

In total, 25 mothers (51%) and 31 fathers (63.3%) had a higher level of education (>10 years of schooling), while 20 mothers and 12 fathers (22.4%) had a lower level of education (<10 years of schooling). Of the sample, 45 children grew up with German as their main language, two were raised to be bilingual and two grew up with Turkish as their main language. Four children (8.2%) had a migration background. Due to this small number, migration background was not considered a possible risk factor in the final analysis.

According to their parents, the children spoke their first word at an average age of 11.8 ± 4.5 months. The girls began speaking at an average age of 10.6 ± 3.8 months, while the boys began speaking at 12.6 ± 4.9 months.

### 3.3. Assessment of Motor Skills

Table 2 presents the results of the motor skills assessment. No significant differences were observed between the boys and the girls.

### 3.4. Language Development Test

All results of the language development test SETK 3–5 are represented in Table 3. The boys scored better than the girls on the subtest ‘Understanding sentences’—they achieved better raw values (*p* = 0.036) and better percentage ranges (*p* = 0.009). There were no significant differences between boys and girls on the other tests.

### 3.5. Leisure Behaviour

The results of the leisure time behaviour are shown in Table 4. On average, the children were physically active for 8.2 ± 6.5 h per week besides the time they spent in kindergarten or at the sports club. The mean time children spent doing sports at the club was 83.0 ± 45.6 min per week. Their average media use time was 154.2 ± 136.2 min per week. The girls watched significantly more television (216.5 ± 157.0 min/week vs. 124.2 ± 82.8 min/week; *p* = 0.015) and had significantly overall more screen time than boys (201.4 ± 168.8 min/week vs. 121.7 ± 99.1 min/week; *p* = 0.043).

### 3.6. Multiple Linear Regression Analysis

In order to examine the influence of individual parameters on the subtests of language development, a linear regression analysis was carried out in each case with the same initial model.

Subcategory ‘Understanding sentences’ (Table 5): The final model explains 87.1% of the variance. Children who demonstrated better flexibility in the SAR subtest (*p* = 0.001) performed better in the language development test US; for each centimetre of increased flexibility, the US raw score increased by 0.051. For each correctly executed jump in the LJ, the US raw score improved by 0.252 (*p* = 0.001). In contrast, the raw score of the subtest US got worse per each centimetre more flexibility reached in SAR (*p* = 0.001) and increased with each additional ground contact in OLS (*p* = 0.012). The more minutes per week children were active in sports clubs, the better their results on the US test (*p* = 0.001), whereas a high amount of time spent in just unstructured activity (beside kindergarten and sports clubs) led to poorer results on the US-test (*p* < 0.001). In the presence of the inactivity risk factor, which was defined according to NAPSE guidelines [35], children scored lower on the US (*p* < 0.001). In contrast, children with the ‘low educational status’ risk factor had better raw scores on the US subtest. The effect of media use time was mixed; while children with overall high weekly media use time showed lower raw scores on the VS of 0.009 per minute (*p* = 0.003), the children’s raw vs. scores improved by 0.062 per minute spent in front of a computer or game console (*p* < 0.001).

Subcategory ‘Formation of morphological rules’ (Table 6): 69.4% of the variance is explained by the final model. Children who were overweight achieved poorer results on the language development test FMR (*p* = 0.001). Children who spent more time being active in a sports club and in their free time achieved significantly higher raw scores on the language test FMR; FMR raw scores increased by 0.127 per minute of activity in sports clubs (*p* = 0.001) and by 2.174 per minute of activity in their free time outside of kindergarten and sports clubs (*p* < 0.001). In addition, children who had the risk factor ‘inactivity according to NASPE’, scored lower on the FMR subtest (*p* = 0.004). Furthermore, the number of correctly formed plurals in the FMR increased with each floor contact less in the OLS; the children formed 0.387 more correct plurals for each fewer ground contact (*p* = 0.004). In contrast, children with less flexibility in SAR and lower speed in the SR scored higher on the FMR test per each centimetre (*p* = 0.003) and each second (*p* = 0.008). The more time children spent in front of televisions as well as computers and game consoles in their free time, the more points they achieved on the FMR test (*p* = 0.021).

Subcategory ‘Phonological working memory for non-words’ (Table 7): The final model explains 97.7% of the variance. The number of correctly repeated words on the PMN increased by 0.803 with each second the children ran faster in the SR (*p* < 0.001) and by 0.168 with each fewer ground contact the children made during the OLS (*p* < 0.001). In contrast, each correctly performed jump at the test LJ resulted in 0.188 fewer points in the PMN (*p* < 0.001). When considering leisure time behaviour, the PMN score improved by 0.042 for each minute of being active in sports clubs per week (*p* < 0.001) and by 0.844 for each minute of activity outside of kindergarten and sports clubs per week (*p* < 0.001). Furthermore, children who showed the risk factors ‘being overweight’, ‘low educational status’ or ‘inactivity according to NAPSE guidelines’ scored significantly lower on the PMN test. Children who spent more time in front of computers and game consoles were able to repeat more words correctly in the PMN test (*p* < 0.001); they achieved 0.042 higher raw scores per minute spent on the computer or game console per week.

## 4. Discussion

This pilot study investigated the influence of various factors related to preschool children’s leisure time behaviour, selected sociodemographic data and selected medical risk factors on their motor skills and linguistic development. In general, this study confirmed the increase in inactivity and media use time in childhood shown in previous studies [1,4,5,6,7,9,30], characterising modern children’s lifestyle. The negative consequences of this inactive lifestyle on children’s health and development are far-reaching and shown in several previous studies [7,17,37], whereas vigorous physical activity was proved to enhance the physical and psychosocial development [25], children’s motor skill abilities [13,18,19,20,21] and decrease the prevalence of overweight [22,23,24]. Additionally, there is broad evidence that being physically active plays an essential role in children’s cognitive development. In several previous studies children who showed a higher frequency of physical activity or better motor skills, achieved better results in individual cognitive development tests [26,27,28,29,30,31,38,39,40]. Additionally, Davis et al. [41] found a positive association between visual processing skills and fine manual control skills.

The possible effects of motor skills on certain areas of language development have not yet been properly researched. So far, correlations between motor skills, language and cognitive development have mostly been investigated when there was an existing pathology in one of these areas [42,43,44,45,46,47,48,49,50,51]. In this study, however, children with speech impediments or any medical impairments were excluded.

In our study, inactive children had poorer scores on all subtests of language development, compared to children who spent more time each week in structured activity at sports clubs. Furthermore, the weekly time of unstructured activity besides kindergarten and sports clubs had a positive influence on the language subtests ‘Formation of morphological rules’ and ’Phonological memory for non-words’. Wang et al. [52] found a possible predictive positive association between the language performance at an early age and the development of later fine and gross motor skills in 11,999 three- to five-year-old children, not showing a significant unique correlation. In contrast to our study, the motor and linguistic development was only recorded by using maternal questionnaires. Our results regarding possible predictive effects of motor skills on language development showed a heterogeneous picture. Test items, which require strength, endurance and speed, in particular, showed no clear correlation with the results of the language development tests. In turn, children who achieved better results on the motor test ‘One-leg stand’ scored higher on the language tests ‘Formation of morphological rules’ and ’Phonological memory for non-words’. Children with better flexibility on the ‘Sit-and-reach’ and better motor skills on the ‘Lateral jumping’ also achieved higher scores in the subcategory ‘Understanding Sentences’. Among the essential requirements of the ‘Sit-and-reach’ and ‘Lateral jumping’ tests are good endurance and flexibility; the ‘One-leg stand’, meanwhile, demands strong coordination and concentration. This ability to concentrate is likewise essential for understanding and correctly performing language development tasks. Thus, cognitive abilities and a high level of concentration are essential for age-appropriate language learning. Language, in turn, is necessary to understand, learn and refine complex motor movements. For example, while a child can learn the general way of performing a one-legged stand by observing other children, he or she can optimise that performance via external linguistic support (e.g., an explanation of how the arms can be used for balance and why).

In contrast, children with a higher total weekly media use time scored higher on the ‘Formation of morphological rules’ but lower on the subtests ‘Understanding sentences’ and ’Phonological memory for non-words’. Anderson et al. [53] previously reported a similar variable effect on children’s cognitive development. While children under two years of age showed negative developmental effects following high media consumption time, preschool-aged children showed varying negative and positive effects depending on their screen time overall, the social context when viewing and the type of programme they watch (e.g., educational programmes for children vs. entertainment programmes for children). In our study, computer and game console time, in particular, was associated with better results on the ’Phonological memory for non-words’ and ‘Understanding sentences’ subtests. Because this study conducted no differentiated recording of the participants’ content-related PC and game console use, the extent to which educational games could have served as a positive influencing factor remains speculative at present. A study by Thorell [54], for example, found that preschool children’s working memory improved after they participated in computer-based training. On the other hand, studies have also linked television-only screen time (i.e., watching TV rather than playing video games) with deficits in language development, a decrease in cognitive performance [53,55,56,57] and an increase in obesity [58].

Increased BMI, in turn, is negatively associated with cognitive and motor development in 3.5- to 7-year-old children [59]. Associations with language development have not yet been investigated. In our analysis, children who were overweight scored significantly lower on the ‘Formation of morphological rules’ and ’Phonological memory for non-words’ language subtests. However, being overweight per se is presumably not a risk factor for poor language development, it is associated with increased television consumption, lack of exercise and a lower level of parental education [60,61]. In our analysis, the social status, as well as the educational level of the parents, was also negatively associated with the language development status of the children. Niemistö [62] and the KiGGS wave 2 [10] also demonstrated these findings a positive correlation between a middle or high socioeconomic status and the motor skills in young children. Better financial status of the parents and growing up in an environment where children have easier access to learning materials seem to promote children’s development.

## 5. Strengths and Limitations

In this pilot study, the relationship between preschool children’s motor skills and language development was further investigated. The consideration of only selected subtests of the SETK 3–5, as well as the small sample, are limiting factors of this study. This must be taken into account when interpreting the results.

Overall, however, the present study found a positive correlation between movement and language in preschool-aged children. The study also highlighted the differentiated influence of screen media; educational games may represent a superior alternative to ‘parking the child in front of the television’.

## 6. Conclusions

Considering all these results, it seems to become more and more important to regard the correlation between all aspects of children’s development and how they affect each other, starting with good cognitive abilities and therefore high concentration skills to be able to develop a greater language ability as a precondition to understand and learn more precise motor coordination movements.

Despite the small sample size, the results of this pilot study serve as a starting point for future studies with larger samples to examine these potential connections between language development and motor skills in childhood. While the small number of participants, especially three-year-old participants, prevented the present study from including all language development tests in its final analysis, future studies should include them to facilitate specific statements about the relationship between individual aspects of language development and motor skills.

However, the results of the present study already underscore the complexity of child development and the close interconnection of cognitive, linguistic and motor skills as well as their great variability of potential positive and negative influencing factors. Therefore, the connection between rising inactivity, increased media consumption, an increased amount of overweight and obese children and the resulting motor deficits should play a more important role as a starting point for future preventive and promotional measures in childhood. These influencing factors—and their connections to motor skills and language development—should also be analysed in greater detail within the framework of larger study groups to further verify the results of this pilot study and possible correlations.

## Figures and Tables

**Figure 1 children-09-00431-f001:**
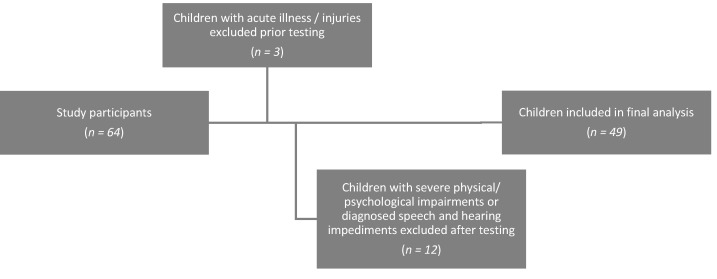
Selection of study participants.

**Figure 2 children-09-00431-f002:**
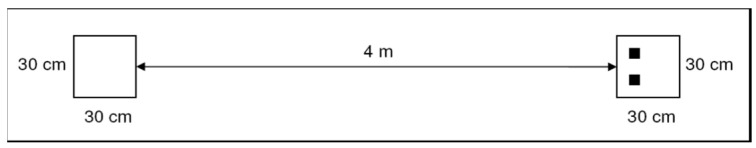
Test setup ‘Shuttle Run’, ©Testmanual KiMo-Test; Dr. rer. nat. S. Dordel, Dr. sports scient. B. Koch, Dipl.-sports scient. D. Klein; Institute of Movement and Neurosciences German Sport University Cologne, Germany.

**Figure 3 children-09-00431-f003:**
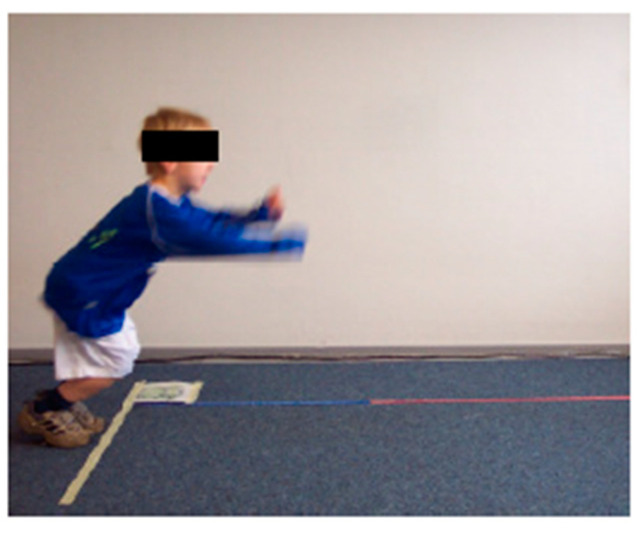
Test setup ‘Standing long jump’, ©Testmanual KiMo-Test; Dr. rer. nat. S. Dordel, Dr. sports scient. B. Koch, Dipl.-sports scient. D. Klein; Institute of Movement and Neurosciences German Sport University Cologne, Germany.

**Figure 4 children-09-00431-f004:**
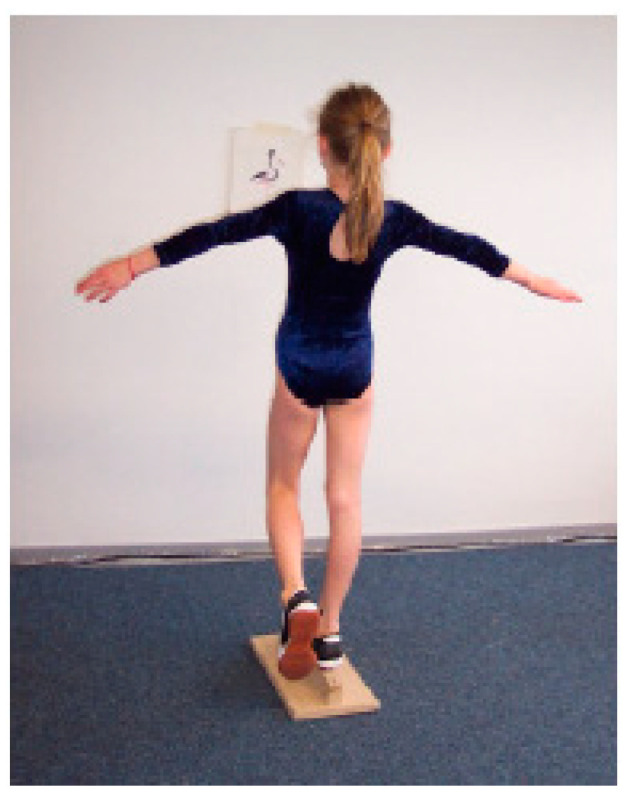
Test setup ‘One-leg stand’, ©Testmanual KiMo-Test; Dr. rer. Nat. S. Dordel, Dr. sports scient. B. Koch, Dipl.-sports scient. D. Klein; Institute of Movement and Neurosciences German Sport University Cologne, Germany.

**Figure 5 children-09-00431-f005:**
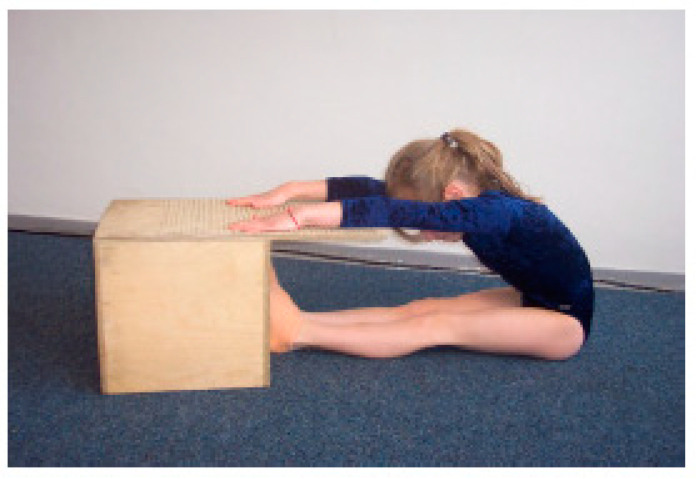
Test setup ‘Sit-and-reach’, ©Testmanual KiMo-Test; Dr. rer. nat. S. Dordel, Dr. sports scient. B. Koch, Dipl.-sports scient. D. Klein; Institute of Movement and Neurosciences German Sport University Cologne, Germany.

**Figure 6 children-09-00431-f006:**
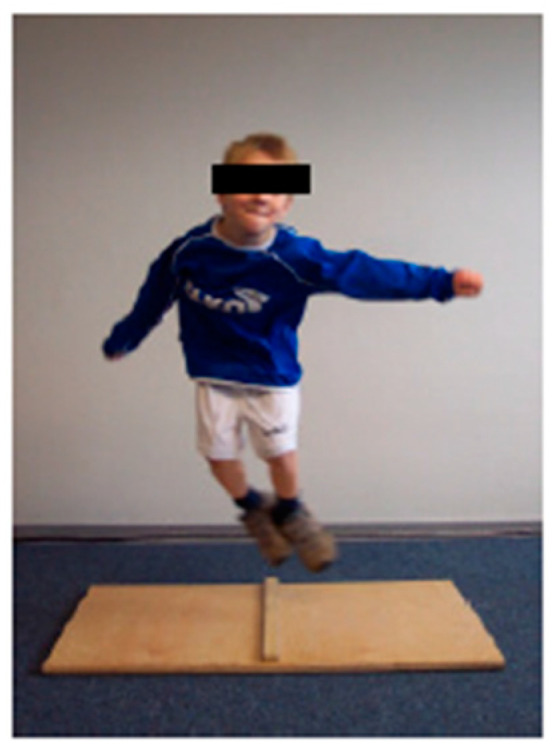
Test setup ‘Lateral jumping’, ©Testmanual KiMo-Test; Dr. rer. nat. S. Dordel, Dr. sports scient. B. Koch, Dipl.-sports scient. D. Klein; Institute of Movement and Neurosciences German Sport University Cologne, Germany.

**Table 1 children-09-00431-t001:** Anthropometric data overall and separated by gender.

	Means ± SD	Min.	Max.	n	*p*-Value *
Age (years)					
all	4.7 ± 0.7	3.3	5.9	49	
girls	4.4 ± 0.7	3.3	5.6	20	0.065
boys	4.8 ± 0.7	3.3	5.9	29	
Height (cm)					
all	108.8 ± 7.1	96.0	124.0	49	
girls	106.5 ± 6.2	96.0	119.0	20	0.063
boys	110.3 ± 7.4	99.0	124.0	29	
Weight (kg)					
all	18.7 ± 2.8	14.1	25.2	49	
girls	18.1 ± 2.5	15.0	23.3	20	0.152
boys	19.2 ± 2.9	14.5	25.2	29	
BMI (kg/m²)					
all	15.8 ± 0.9	14.1	18.3	49	
girls	15.9 ± 1.0	14.7	18.3	20	0.592
boys	15.7 ± 0.9	14.0	17.3	29	

SD = standard deviation, Min. = minimum, Max. = maximum, cm = centimetres, kg = kilogram, * unpaired *t*-test.

**Table 2 children-09-00431-t002:** Results of the motor skills assessment KiMo overall and separated by gender.

	Means ± SD	Min.	Max.	n	*p*-Value *
SR (cm)					
all	11.6 ± 2.6	7.9	20.9	49	
girls	12.0 ± 2.1	7.9	15.6	20	0.375
boys	11.2 ± 2.9	8.0	20.9	29	
SLJ (cm)					
all	84.3 ± 25.9	<0.1	128.0	44	
girls	81.6 ± 23.2	48.0	128.0	16	0.475
boys	85.8 ± 27.6	<0.1	125.0	28	
SAR (cm)					
all	4.5 ± 4.6	−10.0	13.0	48	
girls	6.5 ± 3.9	<0.1	13.0	20	0.779
boys	3.0 ± 4.7	−10.0	12.0	28	
OLS (amount of ground contacts)					
all	19.1 ± 9.7	1.0	31.0	49	
girls	14.5 ± 9.6	1.0	27.0	20	0.393
boys	22.2 ± 8.6	6.0	31.0	29	
LJ (number of jumps)					
all	19.7 ± 7.3	4.0	35.0	48	
girls	17.7 ± 6.4	9.0	31.0	20	0.325
boys	21.2 ± 7.6	4.0	35.0	28	

SR = Shuttle run, SLJ = Standing long jump, SAR = Sit and reach, OLS = One leg stand, LJ = Lateral jumping, SD = standard deviation, Min. = minimum, Max. = maximum, cm = centimetres, * unpaired *t*-test.

**Table 3 children-09-00431-t003:** Results of the **language** development test SETK 3–5 overall and separated by gender.

	Means ± SD	Min.	Max.	n	*p*-Value *
US (RV)					
all	11.7 ± 2.8	2.0	18.0	47	
girls	10.7 ± 3.4	2.0	16.0	19	0.036
boys	12.4 ± 2.0	8.0	18.0	28	
US (PR)					
all	65.0 ± 24.8	0.6	99.2	47	
girls	56.4 ± 30.4	0.6	99.1	19	0.009
boys	70.9 ± 18.6	38.2	98.6	28	
FMR (RV)					
all	22.4 ± 7.6	3.0	36.0	47	
girls	22.2 ± 8.0	9.0	36.0	19	0.543
boys	22.5 ± 7.4	3.0	35.0	28	
FMR (PR)					
all	61.1 ± 30.0	8.1	100.0	47	
girls	64.9 ± 28.3	13.6	100.0	19	0.381
boys	58.5 ± 31.3	8.1	99.9	28	
PMN (RV)					
all	11.5 ± 3.3	5.0	18.0	46	
girls	11.1 ± 3.8	5.0	18.0	19	0.561
boys	11.7 ± 3.0	6.0	17.0	27	
PMN (PR)					
all	66.9 ± 26.2	6.7	99.4	27	
girls	67.9 ± 31.3	6.7	99.4	19	0.074
boys	66.2 ± 22.6	18.4	97.7	46	

US = Understanding of sentences, FMR = Formation of morphological rules, PMN = Phonological working memory for nonwords, RV = raw values, PR = percentages, SD = standard deviation, Min. = minimum, Max. = maximum, * unpaired *t*-test.

**Table 4 children-09-00431-t004:** Means of leisure time behaviour overall and separated by gender.

	Means ± SD	Min.	Max.	n	*p*-Value *
Time spent in sports clubs (min/week)					
all	83.0 ± 45.6	<0.1	225.0	47	
girls	77.1 ± 44.4	<0.1	180.0	19	0.473
boys	87.0 ± 46.8	<0.1	225.0	28	
Physical activity outside of nursery school and sports clubs (h/week)					
all	8.2 ± 6.5	<0.1	31.0	48	
girls	9.9 ± 7.5	<0.1	31.0	20	0.133
boys	7.0 ± 5.4	<0.1	21.0	28	
Media consumption time (min/week)					
all	154.2 ± 136.2	<0.1	600.0	49	
girls	201.4 ± 168.8	<0.1	600.0	20	0.043
boys	121.7 ± 99.1	<0.1	355.0	29	
Time spent watching TV (min/week)					
all	162.0 ± 125.8	22.0	600.0	44	
girls	216.5 ± 157.0	22.0	600.0	18	0.015
boys	124.2 ± 82.8	45.0	330.0	26	
Time spent in front of computer (min/week)					
all	14.8 ± 27.7	<0.1	100.0	29	
girls	9.3 ± 25.6	<0.1	90.0	14	0.307
boys	20.0 ± 29.5	<0.1	100.0	15	

h = hours, min = minutes, SD = standard deviation, Min. = minimum, Max. = maximum, * unpaired *t*-test.

**Table 5 children-09-00431-t005:** Multiple linear regression analysis—End model of the subcategory ‘Understanding sentences’.

	Understanding Sentences			
	Regression Coefficient B	β-Coefficient	*p*-Value	Corrected R²
SLJ (cm)	0.051	−0.553	0.001	0.871
SAR (cm)	0.273	0.519	0.001	
OLS (amount of ground contacts)	0.077	0.334	0.012	
LJ (amount of jumps)	0.252	0.735	0.002	
Risk factor low level of education	−2.865	0.600	0.005	
Weekly time spent in sports clubs (min)	0.040	0.758	0.001	
Weekly physical activity outside of nursery school and sports clubs (h)	−0.286	−0.879	<0.001	
Physical inactivity (by NASPE)	3.832	−0.746	<0.001	
Weekly time spent in front of a computer (min)	0.062	0.818	<0.001	
Weekly media consumption time (min)	−0.009	−0.530	0.003	

SLJ = Standing long jump, SAR = Sit and reach, OLS = One leg stand, LJ = Lateral jumping, NASPE = National Association for Sport and Physical Education, cm = centimetres, min = minutes, h = hours.

**Table 6 children-09-00431-t006:** Multiple linear regression analysis—End model of the subcategory ‘Formation of morphological rules’.

	Formation of Morphological Rules			
	Regression Coefficient B	β-Coefficient	*p*-Value	Corrected R²
Risk factor overweight	22.349	−0.767	0.001	0.694
SR (sec.)	2.246	0.627	0.008	
SAR (cm)	−0.889	−0.633	0.003	
OLS (amount of ground contacts)	−0.387	−0.626	0.004	
Weekly time spent in sports clubs (min)	0.127	0.895	0.001	
Weekly physical activity outside of nursery school and sports clubs (h)	2.174	2.499	<0.001	
Physical inactivity (by NASPE)	9.108	−0.664	0.004	
weekly media consumption time (min)	0.019	0.437	0.021	

SR = Shuttle Run, SAR = Sit and reach, OLS = One leg stand, NASPE = National Association for Sport and Physical Education, sec. = seconds, cm = centimetres, min = minutes, h = hours.

**Table 7 children-09-00431-t007:** Multiple linear regression analysis—End model of the subcategory ‘Phonological working memory for nonwords’.

	Phonological Working Memory for Non-Words			
	Regression Coefficient B	β-Coefficient	*p*-Value	Corrected R²
Risk factor overweight	3.327	−0.215	0.005	0.977
SR (sec.)	−0.803	−0.423	<0.001	
OLS (amount of ground contacts)	−0.168	−0.513	<0.001	
LJ (amount of jumps)	−0.188	−0.387	<0.001	
Risk factor low level of education	3.755	−0.554	<0.001	
Weekly time in sports clubs (min)	0.042	0.532	<0.001	
Weekly physical activity outside of nursery school and sports clubs (h)	0.844	1.829	<0.001	
Physical inactivity (by NASPE)	2.692	−0.370	<0.001	
Weekly time spent in front of computer (min)	0.042	0.388	<0.001	

SR = Shuttle Run, OLS = One leg stand, LJ = Lateral jumping, NASPE = National Association for Sport and Physical Education, sec.= seconds, min = minutes, h = hours.

## Data Availability

The data used and analyzed during the current study involve sensitive patient information and indirect identifiers.

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
