# Peer review of "Correlation between Language Development and Motor Skills, Physical Activity, and Leisure Time Behaviour in Preschool-Aged Children"

_children, 2022, doi:10.3390/children9030431_

Round 1

Reviewer 1 Report

In my opinion, the introduction provides adequate information and structure to set up the research questions raised in the manuscript; the methodology provides sufficient detail, but that can still be an improvement; the results section needs more care in the formatting of the tables and an adjustment to the way in which the final multiple regression models report; the discussion and the conclusion of the study should be improved, trying to be more objective, focusing only on the main contributions of the study.

After carefully reading your manuscript, I point out some aspects that must be improved and corrected:

  1. The title should be adjusted, given that the authors only explored the correlation of language development with the children's lifestyle (or leisure-time behaviour) and their motor performance;
  2. Throughout the manuscript I find several designations for the same concept, for example, motor performance skills, motor performance, motor skills, motor development (keyword), motor ability tests. Authors should try to use the same designation in the title, abstract, keyword and in the body of the manuscript;
  3. The objective of this study is not stated objectively in the abstract (lines 14-15), as well as in the introduction (lines 55-56). Effectively, the authors sought to explore the relationship of language development with hypothetical predictors related to the child's motor performance and lifestyle (physical activity and media use time);
  4. The way the authors report the results in the abstract is not the most correct when applying a multiple linear regression (line 21-27). The authors found a predictor model for each subtest (US, FMR, PMN);
  5. In my opinion, the following sentence in the abstract is too empty needs to be developed further “These results underscore the urgency of preventive 29 measures” (lines 29-30);
  6. The authors did not mention the authorship of the CHILT-III instrument (line 68). A detailed description of the CHILT-III questionnaire (lines 71-82) could be in the testing procedure section. In this section, the authors should try to describe the measurement units of the time variables to better understand table 4. In this table, the same variable is reported in minutes and hours, why?
  7. In line 84, the subtitle "study population" is not appropriate. authors must change to the “sample”. In this section, authors should mention how they recruited a sample (convenience sample?);
  8. In the anthropometric data assessment section (line 98), the authors should mention the anthropometric measurement protocol.
  9. In the results section, the authors should standardize the formatting of the tables and without vertical lines; the legends of the tables must be above the tables; all statistical symbols must be in italics; remove the abbreviations outside the table and put a note below (Note: SR = Shuttle run, SLJ = Standing long jump, SAR = Sit and reach, OLS = One leg stand, LJ = Lateral jumping, cm = centimetres, p-value from t-test)
  10. In table 3, the authors report the same variable in raw values and in percentages, what is the advantage? In lines 234-235, the authors mention " The boys scored better than the girls on the subtest ‘understanding sentences’ (p = 0.009). There were no significant differences between boys and girls on the other tests. ", but in fact when reading the data reported in table 3 the readers can find another statistically significant difference US-RV (p=.036).
  11. In the first line of table 5, 6 7, it is suggested that the abbreviations (US, FMR, PMN) be written in full for a better understanding of the reader;
  12. In lines 357 and 358, the authors write the following: “In this pilot study, the relationship between preschool children’s motor performance and language development was investigated for the first time”. This sentence should be reformulated as it is not true given what it says in the introduction (line 52) and there are other studies as an example

Wang, M.V., Lekhal, R., Aaro, L.E. et al. The developmental relationship between language and motor performance from 3 to 5 years of age: a prospective longitudinal population study. BMC Psychol 2, 34 (2014). https://doi.org/10.1186/s40359-014-0034-3

  1. All statistical symbols must be in italics (n, p, r…. ). 
  2. Some aspects of formatting should be corrected (spelling, punctuation). Please, correct what is pointed out in the body of the manuscript;

Author Response

Dear editor,

we would like to express our gratitude for reviewing our manuscript again. Your comments greatly helped us improve the quality of the paper. Please find our point-to-point answers to your comments below in italics. We hope that we have been able to address the concerns and that the manuscript has now gained in quality.

Best wishes,

Daniela Mulé in the name of all authors

Reviewer 1

Comments and Suggestions for Authors

In my opinion, the introduction provides adequate information and structure to set up the research questions raised in the manuscript; the methodology provides sufficient detail, but that can still be an improvement; the results section needs more care in the formatting of the tables and an adjustment to the way in which the final multiple regression models report; the discussion and the conclusion of the study should be improved, trying to be more objective, focusing only on the main contributions of the study.

After carefully reading your manuscript, I point out some aspects that must be improved and corrected:

1. The title should be adjusted, given that the authors only explored the correlation of language development with the children's lifestyle (or leisure-time behaviour) and their motor performance;

Thank you for your comment. The title has been adjusted accordingly.

2. Throughout the manuscript I find several designations for the same concept, for example, motor performance skills, motor performance, motor skills, motor development (keyword), motor ability tests. Authors should try to use the same designation in the title, abstract, keyword and in the body of the manuscript;

We absolutely agree. The terminology has been harmonized accordingly.

3. The objective of this study is not stated objectively in the abstract (lines 14-15), as well as in the introduction (lines 55-56). Effectively, the authors sought to explore the relationship of language development with hypothetical predictors related to the child's motor performance and lifestyle (physical activity and media use time);

Thank you. The relevant sections in the abstract and in the introduction have been revised and formulated objectively.

4. The way the authors report the results in the abstract is not the most correct when applying a multiple linear regression (line 21-27). The authors found a predictor model for each subtest (US, FMR, PMN);

Thank you for this important advice. Further explanations have been added and sentences have been adjusted to clarify that these are "predictor models". The problem is that the number of words is limited to 200.

5. In my opinion, the following sentence in the abstract is too empty needs to be developed further “These results underscore the urgency of preventive measures” (lines 29-30);

We also thank you for this comment. The sentence was adjusted to clarify the key message of the "Conclusion". 

6. The authors did not mention the authorship of the CHILT-III instrument (line 68). A detailed description of the CHILT-III questionnaire (lines 71-82) could be in the testing procedure section. In this section, the authors should try to describe the measurement units of the time variables to better understand table 4. In this table, the same variable is reported in minutes and hours, why?

Thank you for your comment. The literature was supplemented accordingly and the description of the modified CHILT III questionnaire was added. For a better overview, the results were presented uniformly in minutes, with the exception of the total activity.

7. In line 84, the subtitle "study population" is not appropriate. authors must change to the “sample”. In this section, authors should mention how they recruited a sample (convenience sample?);

We also agree with this comment. The headline was changed accordingly and the recruitment of the study participants was described.

8. In the anthropometric data assessment section (line 98), the authors should mention the anthropometric measurement protocol.

We would also like to thank you for this advice. The recording of anthropometric data in a measurement protocol is now mentioned in the corresponding section.

9. In the results section, the authors should standardize the formatting of the tables and without vertical lines; the legends of the tables must be above the tables; all statistical symbols must be in italics; remove the abbreviations outside the table and put a note below (Note: SR = Shuttle run, SLJ = Standing long jump, SAR = Sit and reach, OLS = One leg stand, LJ = Lateral jumping, cm = centimetres, p-value from t-test)

We absolutely agree. The table format has been standardised, the statistical symbols have been formatted in italics, the table headings have been moved above the tables and the abbreviations have been moved to a footnote below each table.

10. In table 3, the authors report the same variable in raw values and in percentages, what is the advantage? In lines 234-235, the authors mention " The boys scored better than the girls on the subtest ‘understanding sentences’ (p = 0.009). There were no significant differences between boys and girls on the other tests. ", but in fact when reading the data reported in table 3 the readers can find another statistically significant difference US-RV (p=.036).

Thank you very much for this important information. The significantly better performance of the boys in the US test, both in the raw scores and the adjusted percentile ranks, has now been presented in a more differentiated way.

11. In the first line of table 5, 6 7, it is suggested that the abbreviations (US, FMR, PMN) be written in full for a better understanding of the reader;

Very correct. The abbreviations of the language tests have been written out in the tables accordingly.

12. In lines 357 and 358, the authors write the following: “In this pilot study, the relationship between preschool children’s motor performance and language development was investigated for the first time”. This sentence should be reformulated as it is not true given what it says in the introduction (line 52) and there are other studies as an example

Wang, M.V., Lekhal, R., Aaro, L.E. et al. The developmental relationship between language and motor performance from 3 to 5 years of age: a prospective longitudinal population study. BMC Psychol 2, 34 (2014). https://doi.org/10.1186/s40359-014-0034-3

            Thank you very much. we have added these and other studies to the discussion.

13. All statistical symbols must be in italics (n, p, r…. ). 

Absolutely correct. The statistical symbols have been formatted in italics.

14. Some aspects of formatting should be corrected (spelling, punctuation). Please, correct what is pointed out in the body of the manuscript;

Many thanks also for this hint. The formatting and spelling errors have been corrected accordingly.

Reviewer 2 Report

The issue raised in the paper is interesting in terms of the child's development. The construction of the manuscript is quite good, however, I suggest some corrections:

The authors' names should be sorted out.

Descriptions of the Tables should be placed above the Tables.

Do you have any illustrations/photos of the performed tests?

I suggest replacing Table 3 with Figure.

I feel that the Discussion is not deep enough, please compare more studies and analyze your findings.

In the text, you use so many abbreviations, that it is hard to read the manuscript smoothly.

Best regards

Author Response

Dear editor,

we would like to express our gratitude for reviewing our manuscript again. Your comments greatly helped us improve the quality of the paper. Please find our point-to-point answers to your comments below in italics. We hope that we have been able to address the concerns and that the manuscript has now gained in quality.

Best wishes,

Daniela Mulé in the name of all authors

Reviewer 2

Comments and Suggestions for Authors

The issue raised in the paper is interesting in terms of the child's development. The construction of the manuscript is quite good, however, I suggest some corrections:

The authors' names should be sorted out.

Thank you very much for your comment. We have revised the authors again.

Descriptions of the Tables should be placed above the Tables.

Thank you very much. The table caption has been moved above the table.

Do you have any illustrations/photos of the performed tests?

Thank you for this very valuable suggestion. Corresponding pictures for the motor skills tests have been added.

I suggest replacing Table 3 with Figure.

Many thanks for this suggestion. For a better overview, table 3 has been left as it is, but tables 1-4 have been formatted and made more readable.

I feel that the Discussion is not deep enough, please compare more studies and analyze your findings.

We absolutely agree with the expert and have completely revised and deepened the discussion again.

In the text, you use so many abbreviations, that it is hard to read the manuscript smoothly.

Thank you for this important comment. For better readability, some of the abbreviations (especially the tests) have been written out.
